# A 1-μm-Band Injection-Locked Semiconductor Laser with a High Side-Mode Suppression Ratio and Narrow Linewidth

**DOI:** 10.3390/s22239239

**Published:** 2022-11-28

**Authors:** Jia-Qi Chen, Chao Chen, Qi Guo, Li Qin, Jian-Wei Zhang, Hang-Yu Peng, Yin-Li Zhou, Jing-Jing Sun, Hao Wu, Yong-Sen Yu, Yong-Qiang Ning, Li-Jun Wang

**Affiliations:** 1State Key Laboratory of Luminescence and Application, Changchun Institute of Optics, Fine Mechanics and Physics, Chinese Academy of Sciences, Changchun 130033, China; 2Center of Materials Science and Optoelectronics Engineering, University of Chinese Academy of Sciences, Beijing 100049, China; 3Xiongan Innovation Institute, Chinese Academy of Sciences, Xiongan 071800, China; 4State Key Laboratory of Integrated Optoelectronics, College of Electronic Science and Engineering, Jilin University, Changchun 130012, China

**Keywords:** semiconductor laser, narrow linewidth, side-mode suppression ratio, apodized fiber Bragg grating, frequency noise

## Abstract

We demonstrate a narrow-linewidth, high side-mode suppression ratio (SMSR) semiconductor laser based on the external optical feedback injection locking technology of a femtosecond-apodized (Fs-apodized) fiber Bragg grating (FBG). A single frequency output is achieved by coupling and integrating a wide-gain quantum dot (QD) gain chip with a Fs-apodized FBG in a 1-μm band. We propose this low-cost and high-integration scheme for the preparation of a series of single-frequency seed sources in this wavelength range by characterizing the performance of 1030 nm and 1080 nm lasers. The lasers have a maximum SMSR of 66.3 dB and maximum output power of 134.6 mW. Additionally, the lasers have minimum Lorentzian linewidths that are measured to be 260.5 kHz; however, a minimum integral linewidth less than 180.4 kHz is observed by testing and analyzing the power spectra of the frequency noise values of the lasers.

## 1. Introduction

One-micrometer band high-power fiber lasers exhibiting excellent beam qualities and high operational efficiencies have enabled many technologies that are now widely applied in scientific research, industrial processing, and other fields [1]. The lasers usually exhibit a main oscillation power amplification (MOPA) structure [2,3] to amplify the multistage energy of a single-frequency seed source; the power output in kilowatts can be achieved through the spectral beam combining of the lasers [4,5]. Changes in the output characteristics occur during amplification, such as the mode, linewidth, polarization, and wavelength. Hence, the seed source needs to ensure a narrow linewidth laser output and achieve the side-mode suppression ratio (SMSR) as high as possible to suppress the background noise sidelobe during multistage energy amplification [6]. Series of 1-μm band lasers are needed as seed sources to achieve spectral beam combining [4,5]. The characteristics of the seed sources, such as their spectra, beam qualities, and power levels, dictate the effects of spectral beam combining. Therefore, it is necessary to achieve linewidth compression and noise suppression of the laser in the range of the 1-μm band.

In recent years, the schemes commonly used to achieve 1-μm band lasers mainly include solid-state lasers [7,8], fiber lasers [9,10], and semiconductor lasers [11,12,13,14,15,16,17,18,19,20,21,22]. Although solid-state lasers and fiber lasers can achieve linewidth on the order of kHz, solid-state lasers are large, are sensitive to shock and vibration, have relatively low stability and require additional laser diode (LD) pumping. Fiber lasers also require LD pumping, and the wavelength range is limited by the amplified spontaneous emission (ASE) spectrum of the gain fiber, while the power consumption, size, cost, and noise are high. However, 1-μm band semiconductor lasers can effectively avoid the above problems. Through a reasonable laser structure design, the lasers possess unique advantages, such as a compact structure, reliability, and photoelectric efficiency; additionally, these lasers perform comparably to fiber lasers, especially concerning their linewidth and noise behaviors [11]. Semiconductor lasers with distributed feedback (DFB) and distributed Bragg reflection (DBR) structures [12,13,14,15], through the optimization of their waveguides, grating structures, optical field limiting factors, linewidth broadening factors and transmission control losses, can exhibit linewidth output levels on the order of MHz or less; however, they need complex extension and device structures, which increases the technology complexity and results in high costs. Common external cavity laser (ECL) structures, such as etalons, volume gratings and diffraction gratings [16,17,18], can significantly narrow linewidth, but the low integration levels of lasers and their sensitivities to environmental vibrations affect their reliability and stability characteristics. In contrast, integrated ECLs are popular because of their compact and stable cavity structures with extremely low intensity noise levels; their phase-frequency noise levels can be further suppressed by external cavity feedback technology [19,20,21,22]. However, high integration indicates high process complexity and cost. A series of lasers are required for spectral beam combining, and we need a low-cost scheme that can mass-produce lasers without sacrificing their performance. With the continuous maturation and perfection of semiconductor gain chip and fiber Bragg grating (FBG) fabrication technology, FBG-based ECL has become an interesting candidate process.

In this work, we fabricate narrow linewidth semiconductor lasers based on FBG external optical feedback injection locking technology, which is developed by using a wide gain spectrum quantum dot (QD) gain chip coupled with a femtosecond-apodized (Fs-apodized) FBG. This scheme is highly integrated and inexpensive; the wide gain spectrum of the QD gain chip can realize lasers with more wavelengths that are needed to increase the spectral beam range. The apodized FBG is prepared flexibly by femtosecond laser point-by-point technology. The high Q value of the equivalent resonator is used to realize the main mode selection and to maximize the gain, while the laser achieves linewidth compression and noise suppression by an optical negative feedback mechanism. The laser achieves a maximum SMSR of 66.3 dB and a maximum output power of 134.6 mW. The Lorentzian linewidth of 260.5 kHz and the integral linewidth of 180.4 kHz are obtained by testing and analyzing the power spectra of the frequency noise values of the lasers. It is shown that the scheme can achieve more wavelength seed source lasers throughout the gain spectrum range by characterizing the laser performance at different gain within the gain spectrum of the chip according to the requirements of the seed source and the performance of the gain chip, we include initial results at both 1030 nm and 1080 nm in this paper. The performance of the laser is further improved by optimizing the structural parameters of the chips and gratings.

## 2. Experiment Process

The device structure of a narrow linewidth semiconductor laser based on self-injection locking is summarized in Figure 1a. The QD chip can achieve a wide gain range. While the FBG extends the effective length of the resonator, improves the quality factor Q of the resonator, and plays a role in frequency selection and filtering. The main mode selection and gain maximization are realized by the high Q value of the equivalent Fabry–Pérot (F–P) resonator. The length of equivalent F–P resonator (L_ext_) is 5 mm, which is equivalent to the beam path from the front facet of the gain chip to the reflection center of the FBG in Figure 1d. The light entering the equivalent resonant cavity returns to the laser chip after being fedback by the narrow reflection band gap of the FBG, making the light of this mode more competitive. The carrier density in the cavity changes, and the refractive index of the gain medium also changes. Thus, the oscillation of the selected mode is gradually strengthened, and the threshold is reduced, as shown in Figure 1e. The 3 dB bandwidth of the external cavity FBG reflection spectrum needs to be less than twice the longitudinal mode interval, to ensure the single longitudinal mode laser output.

The Fs-apodized FBG is prepared by combining a half wave plate (HWP) and a polarization beam splitter (PBS) [23,24] to achieve real-time adjustment of femtosecond laser pulse energy. To maintain a high SMSR after multistage power amplification, background noise sidelobes need to be suppressed, and the apodized FBG is designated as the frequency selection element. The femtosecond laser direct writing method [25,26] can more flexibly fabricate apodized FBGs with different apodization function distributions than the phase mask and holographic interference methods. Figure 1b is an example to demonstrate the flexibility of femtosecond laser energy regulation method and prepare a small segment of apodized refractive index modulation grating structure. It can be observed that the obvious morphological gradient in the vision of microscope field. Figure 1c shows the grating structure of the laser used, and it is the refractive index modulated topography in the middle of the FBG. Because the grating is long, it is impossible to observe an obvious morphological gradient in the vision of microscope field. The femtosecond laser used in this experiment is produced by the Light Conversion company, Vilnius, Lithuania; the laser wavelength is 515 nm, the pulse width is 290 fs, and the fiber used is an HI 1060 fiber produced by the Corning Company, Corning, NY, USA. When preparing Fs-apodized FBGs, the stepping motor is controlled by a program to adjust the rotation angle and speed of the HWP, and the laser pulse energy is controlled in real time so that the amplitude of the refractive index modulation conforms to the distribution of the apodization function. The phase matching condition of the FBGs is $m\lambda_{B}=2n_{eff}\Lambda$, where *m* is the grating order, *λ_B_* is the wavelength, *n_eff_* is the effective refractive index and *Λ* is the grating period. A 1030 nm third-order Fs-apodized FBG is prepared with a grating period of 1.064 µm and a length of 6 mm. Figure 2a shows the reflection and transmission spectra of Fs-apodized FBGs with different central wavelengths prepared in a 1-μm band. The full-width at half-maximum of the prepared Fs-apodized FBGs is less than 80 pm, and the SMSR is more than 28 dB. The FBGs adopt a low reflectivity structure with a reflectivity of 30% to maintain the output power of the lasers.

This scheme can achieve many different wavelength single frequency laser seed sources and increase the range of spectral beam combination by using a QD chip with a wide gain spectrum [27,28]; the length L of the chip is 1.5 mm, and the ridge width w is 5 μm. The ridge waveguide is bent by 7° near the front end to eliminate the influence of the cavity effect. The chip has a specified high reflectivity (HR) of 90% at the rear facet to abate the loss in the resonator, reduce the threshold and increase the output power, while the anti-reflectivity (AR) coating at the front facet has a value less than 0.01% to minimize the influence of the internal cavity mode of the chip. There are 4 layers of the AR coating, and the materials are Al_2_O_3_ and TiO_2_. The ASE spectrum of the gain chip is tested by an optical spectrum analyzer (OSA) (AQ6370B, Yokogawa, Tokyo, Japan) with a resolution of 0.02 nm, as shown in Figure 2b. The peak of the gain spectrum blueshifts from 1050 nm to 980 nm by approximately 70 nm as the injection current increases, the gain peak tends to be stable without a lasing peak, and the gain at 1030 nm is always higher than that at 1080 nm. The voltage and output power versus injection current (power−current−voltage (P−I−V)) characteristics of the gain chip are shown in Figure 2c; the figure shows that the lasers can achieve a power output of nearly 30 mW at 400 mA.

We have carried out standard butterfly packaging to accurately calibrate the performance of the laser. The gain chip, FBG and thermistor are fixed on the upper surface of the heat sink, and the heat sink is welded to the upper surface of the thermoelectric cooler (TEC). The fiber front end is polished into a tapered fiber lens; that is, a tapered fiber lens is used to couple the QD gain chip to the FBG. This process simplified the laser housing structure for isolation from the environment. Weld the electrode of the gain chip, thermistor, and TEC with the internal lead of the pin of the shell. Finally, the prepared semiconductor laser is packaged in a standard butterfly housing, which enables us to construct a hermetically sealed enclosure and achieve vibration isolation, acoustic isolation, and thermal isolation of the laser.

## 3. Results

All measurements are performed using a low-noise current source (LDX−3620B, ILX lightwave, Newport, Wuxi, China) and a laser diode temperature controller (LDT−5910C, ILX lightwave). All measurements are conducted at 20 °C. To further suppress the current noise, a low-pass filter (LNF−320, ILX lightwave) is used in the experiment.

### 3.1. Laser Spectra and P−I−V Characteristics

The continuous wave (CW) spectrum characteristics are measured using an OSA. Figure 3 shows the laser spectra. The 1030 nm laser reaches its highest SMSR of 66.3 dB at 1030.098 nm with a 400 mA current. The spectrum at 50 mA shows the transmission peak of the FBG, which is basically consistent with Figure 2a. For the 1080 nm laser, the highest SMSR of 61.8 dB is reached at 1080.444 nm with a current of 400 mA. Both lasers achieve stable single-mode characteristics. There is a wavelength difference between the laser in Figure 3 and the FBG in Figure 2a, which is due to the temperature difference between the grating and the laser during the measurement process, as well as the stress on the grating when the laser is packaged.

From the measured P–I–V curve shown in Figure 4a, we determine the threshold current and the slope efficiency to be 70 mA and 0.40 W/A for the 1030 nm laser, respectively, while the maximum output power is 134.6 mW at 400 mA. Figure 4c,e show the changes in the laser spectra versus the changes in the injection currents for the 1030 nm laser. The wavelength tuning rate is 1.13 nm/A, and the continuous wavelength tuning range is 28.2 pm. The spectrums have a mode hopping phenomenon, which is consistent with the P–I–V curves. The 1030 nm laser maintains an SMSR of more than 60 dB under currents above 150 mA and that is more than 45 dB for the 1080 nm laser at the same range. The 15 dB minimum SMSR difference between the two lasers (>150 mA) is due to the gain at 1030 nm being higher than that at 1080 nm, which is closer to the peak value of the ASE spectrum. This phenomenon is also the reason for the difference in the threshold currents mentioned later. In addition, the threshold current of the 1080 nm laser is 144 mA, and the maximum output power is 133.0 mW with a slope efficiency of 0.51 W/A in Figure 4b. The wavelength tuning rate is 1.18 nm/A, and the continuous wavelength tuning range is 45.3 pm in Figure 4d,f. The difference in the threshold current is due to the difference in gain. The continuous tuning wavelength difference between the two lasers is due to the slight deviation of the fiber position during the artificial cleavage process, which changes the cavity length of the laser. We can reduce the energy loss and increase the continuous tuning wavelength of the laser by optimizing the structural parameters of the fiber grating in future work.

### 3.2. Polarization Characteristics

Previous studies have shown that femtosecond laser point-by-point writing FBG has birefringence [26,29], which may lead to linearly polarized laser output. To verify this phenomenon, the polarization extinction ratio (PER) of the laser is characterized. A linear polarization controller (Thorlabs, FBR-LPNIR), adjustable fiber-to-fiber coupler (Thorlabs, FBP-C-FC) and power detector are used to test the polarization characteristics of the lasers. By carefully rotating the polarization controller for one cycle and recording the change in power passing through the polarization controller, a polar diagram of the output power and rotation angle is obtained, as shown in Figure 5a. The measured maximum power *P_max_* and minimum power *P_min_* are 113.56 mW and 1.91 mW under 400 mA, which give a PER value exceeding 17.74 dB. ($\mathrm{PER}=10log_{10}\left(P_{max}/P_{min}\right)$). The PER value at different currents is shown in Figure 5b. The TE and TM modes are not lasing at 50 mA which is below the threshold current. The difference of the power level between TE mode and TM mode is small, and the PER is low. The maximum PER value is 18.68 dB. It should be noted that the laser output fiber is a single-mode fiber, but it still shows linear polarization performance. This is due to the birefringence characteristics introduced by the femtosecond laser writing asymmetric refractive index modulation structure. Higher PER probably obtained if Fs-apodized FBG is written in the polarization-maintaining fiber, which is also part of the subsequent work.

### 3.3. Linewidth and Relative Intensity Noise Characteristics

Furthermore, to understand the change process of frequency noise affecting the laser linetype and observe the distribution characteristics of linewidth under different Fourier frequency, we chose the phase noise power spectral density (PSD) of the laser [30] to derive linewidth instead of the commonly used self-heterodyne interferometry. The laser output adopts a 120° phase difference interferometer [31] with a 3 × 3 coupler, and the differential phase information is measured. The light passing through output ports 1 and 2 of the coupler is reflected back to the coupler by the Faraday mirror and interferes with the reference light in the delay fiber previously split. After the output data are collected by an analogue card, the frequency/phase characteristics deviating from the linear scan are analyzed from the linear fitting curve. Finally, the phase noise PSD of the laser is obtained by demodulating the measured phase information.

The frequency noise of the laser can be obtained through the phase noise PSD. The following is the calculation formula between the two:(1)Sбv(f)[Hz2Hz]=2f2L(f)
where *S*_бv_(*f*) is the single-sideband PSD of the frequency noise level, *L*(*f*) is the PSD of phase noise and *f* is the Fourier frequency.

Two kinds of noise directly determine the frequency linewidth of the laser. The frequency linewidth is evaluated by integrating the PSD of the frequency noise level using the *β* isolation line method. According to the Wiener–Khintchine theorem, the laser line can be decided by the Fourier transform of the optical field autocorrelation function, that is, the Lorentzian and Gaussian lines. The demarcation point for the two lines in the frequency noise PSD *S*_бv_(*f*) is evaluated by the β isolation line:(2)$$S_{бv}\left(f\right)=8\ln2f/\left(\pi^{2}\right)$$

In the slow frequency region where *S*_бv_(*f*) is higher than the β-separation line, *S*_бv_(*f*) corresponds to a noise level of 1/*f*; that is, *S*_бv_(*f*) > 8ln(2) *f*/(π^2^). The laser power spectrum is a Gaussian-type spectrum [32,33], and *S*_бv_(*f*) directly determines the frequency line width [33]. Therefore, the laser linewidth Δ*v* is obtained by integrating the noise component area A near the intersection of *S*_бv_(*f*) and the β-separation line, which bears the following equations:(3)$$\Delta v_{\mathrm{integral}}={\left(8\ln2A\right)}^{\frac{1}{2}},A=\int_{\frac{1}{T_{0}}}^{\infty }\left[S_{бv}\left(f\right)-8\ln2f/\left(\pi^{2}\right)\right]S_{бv}\left(f\right)df$$
where *H*(*X*) is the unit step function, which has either 0 (*X* > 0) or 1 (*X* < 0).

In contrast, in the fast frequency region where *S*_бv_(*f*) is lower than the β-separation line, the laser power spectrum does not contribute to the linewidth. The white noise and the corresponding Lorentzian linewidth are determined by a modified Schawlow–Townes formula:(4)$$\Delta v_{\mathrm{Lorentz}}=\pi h_{0}$$
where *h*_0_ is the white noise platform.

Figure 6a shows the PSD of a 1030 nm semiconductor laser, and Figure 6b evaluates the laser linewidth in the full Fourier frequency range by integrating the frequency noise fluctuation PSD based on the β isolation line method. The white noise platform is 82,960 Hz^2^/Hz@2 MHz for the 1030 nm laser, the corresponding Lorentzian linewidth is 260.5 kHz, and the minimum integral linewidth is 180.4 kHz. The minimum Lorentzian and integral linewidth of the 1080 nm laser are 309.3 kHz and 208.4 kHz, respectively. The difference in the minimum linewidth between the two lasers is affected by the cleavage of the fiber. The cleavage of the fiber is an interesting research direction for narrowing the laser linewidth in future studies. In addition, the integrated linewidth obtained at the intersection of the β-separation line and the frequency noise spectrum is close to the Lorentzian linewidth; the β-separation line does not consider the Fourier limit and gives very optimistic results in the case of the shortest time [34].

A photodiode, radio-frequency (RF) amplifier, and spectrum analyzer are used to detect and process the laser signals for low− and high−frequency measurements. The relative intensity noise (RIN) PSD of the 1030 nm laser is shown in Figure 7. At a low frequency of 1 kHz, RIN fluctuates from −137 dBc/Hz to −143 dBc/Hz with increasing frequency. In a high frequency range of approximately 1 MHz, the RIN is generally stable between −151 dBc/Hz and −154 dBc/Hz. The low- and high-frequency RIN values of the 1080 nm laser are approximately −141 dBc/Hz and −153 dBc/Hz, respectively. The lasers exhibit the characteristics of 1/f noise below 1 MHz. The RIN above 1 MHz is basically unchanged. The peaks observed in the 1/f noise region are mainly caused by the ambient acoustic and technical noise levels of the driving circuit.

## 4. Conclusions

In this work, we demonstrate a semiconductor laser based on FBG external optical feedback injection locking technology. The wide gain spectrum of the QD gain chip achieves the higher wavelengths of lasers needed to increase the spectral combination range. The Fs-apodized FBGs in a 1-μm band are prepared flexibly by femtosecond laser point-by-point technology. The equivalent F–P resonator is constructed by performing a coupling integration of the external cavity, and the longitudinal mode of the laser is locked on the high Q-factor transmission peak of the equivalent F–P resonator. The threshold current is effectively reduced, and the output power increases. The laser achieves a Lorentzian linewidth of less than 260.5 kHz and a maximum side-mode suppression of 66.3 dB at 1030 nm and 1080 nm; the maximum output power reaches 134.6 mW, and the PER exceeds 18.68 dB. This scheme meets the requirements of spectral beam combining and multistage energy amplification for a 1-μm band high-power fiber laser seed source. The laser has a high SMSR, narrow linewidth, and low fabrication cost, indicating that it can be commercialized. The scheme has important applications in space laser communication, gravitational wave detection, atomic physics and high-energy laser preparation, and it is an ideal seed source for high-power fiber lasers. This scheme can be extended to lasers in other bands, such as C band. Such lasers can be applied to laser radar and fiber distributed sensors.

## Figures and Tables

**Figure 1 sensors-22-09239-f001:**
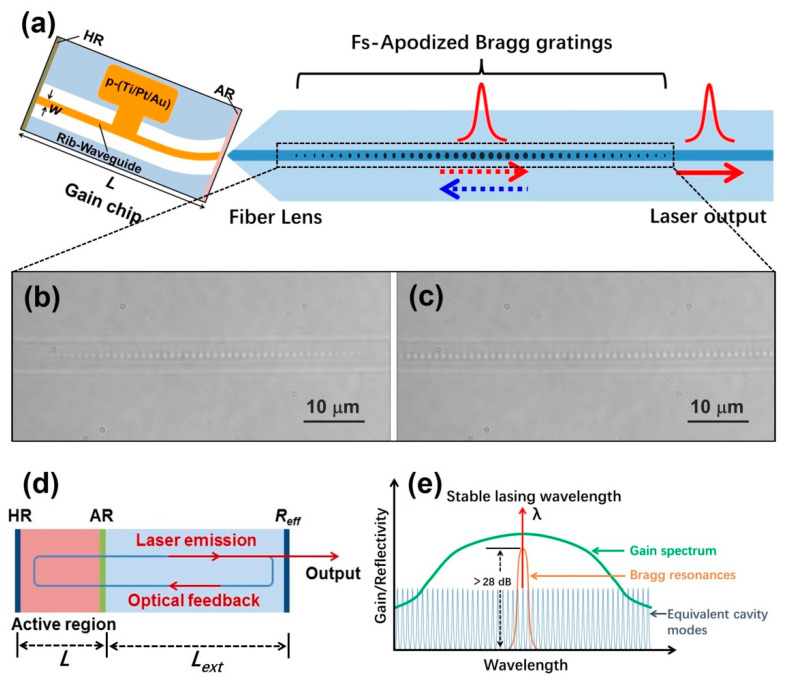
(**a**) Structure diagram of the semiconductor laser. (**b**) Example of the Fs-apodized FBG with different refractive index distributions prepared by energy modulation. (**c**) Microscopy of the Fs-apodized FBG in the middle of the fiber. (**d**) Working principle diagram of the equivalent resonator. (**e**) Schematic diagram of the fiber grating frequency selection.

**Figure 2 sensors-22-09239-f002:**
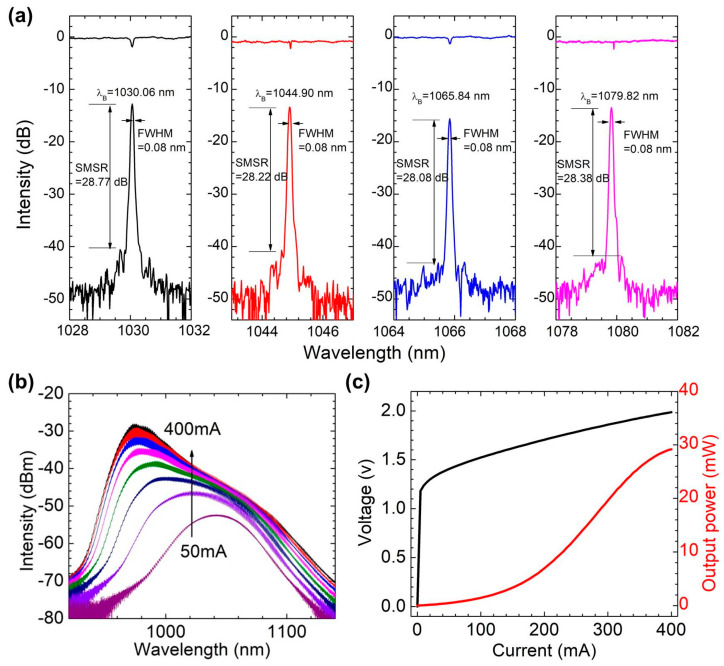
(**a**) Reflection and transmission spectra of Fs-apodized FBGs with central wavelengths of 1030 nm, 1045 nm, 1066 nm and 1080 nm. (**b**) ASE spectrum of the gain chip. (**c**) P−I−V curve of the gain chip.

**Figure 3 sensors-22-09239-f003:**
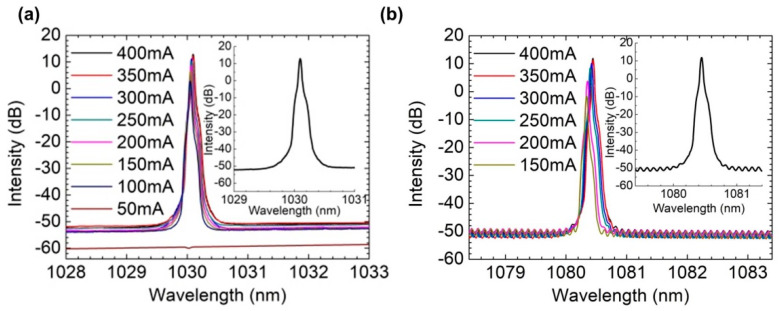
Laser spectrum of the (**a**) 1030 nm and (**b**) 1080 nm lasers under different currents. The inset shows the laser spectrum of the corresponding lasers under a current of 400 mA.

**Figure 4 sensors-22-09239-f004:**
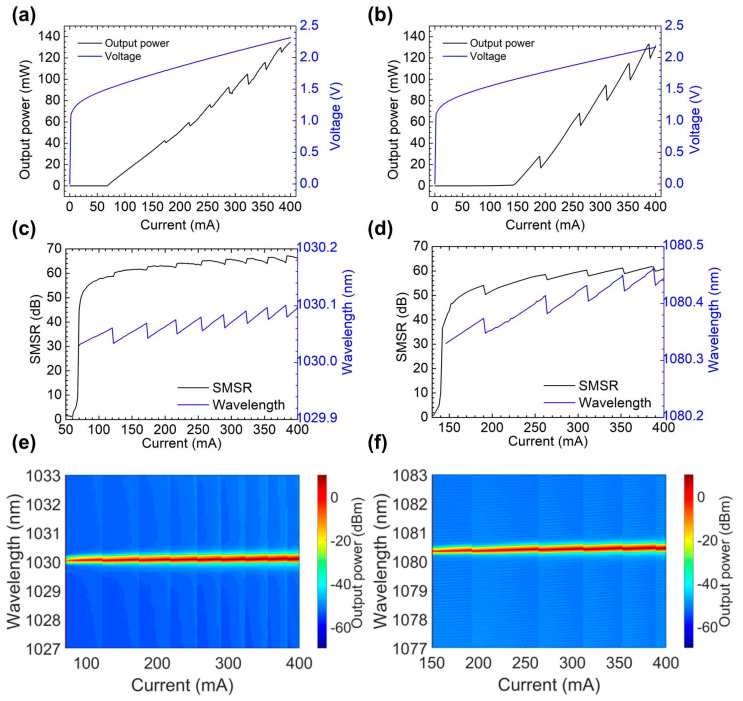
The P−I−V curves of the (**a**) 1030 nm and (**b**) 1080 nm lasers. (**c**,**d**) show the changes in wavelength and SMSR versus the injection currents. (**e**,**f**) are the jet color maps of the laser spectra with injection currents.

**Figure 5 sensors-22-09239-f005:**
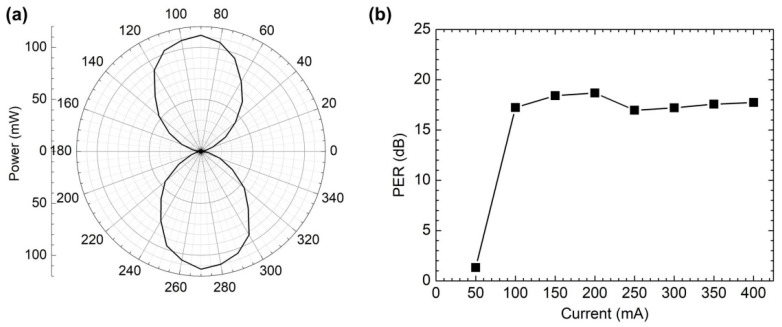
(**a**) The power versus angle of the 1030 nm laser under 400 mA. (**b**) The PER value at different currents.

**Figure 6 sensors-22-09239-f006:**
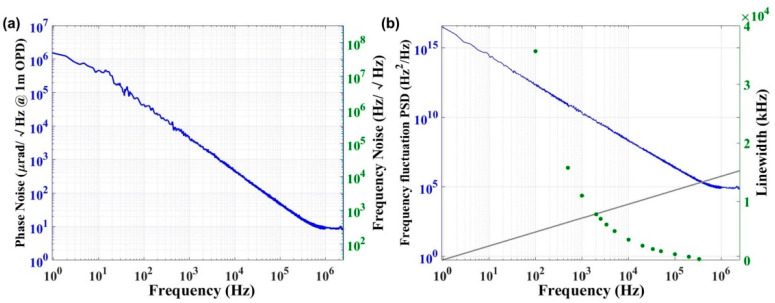
(**a**) Phase noise spectrum of the 1030 nm laser under 246 mA. (**b**) The frequency linewidth of the 1030 nm laser was evaluated by frequency noise PSD integration.

**Figure 7 sensors-22-09239-f007:**
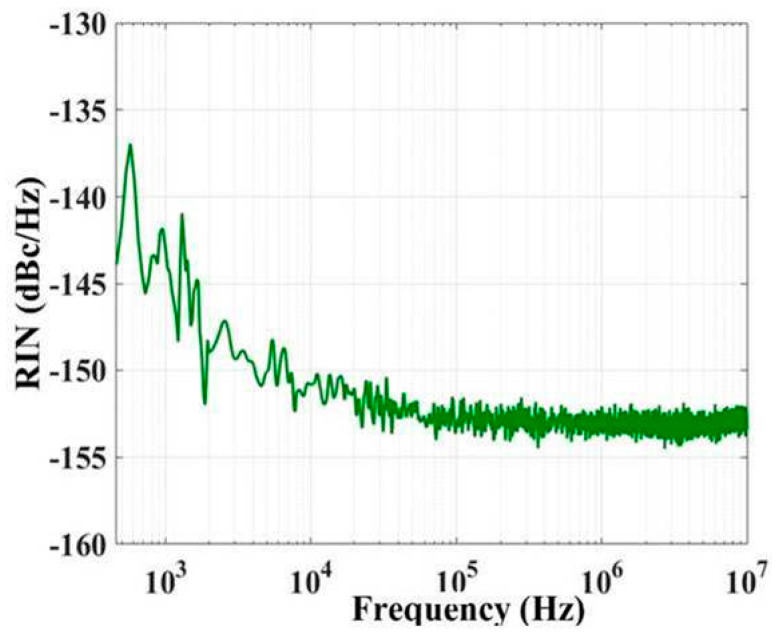
Measured RIN spectrum of the 1030 nm laser measured under 315 mA.

## Data Availability

Data underlying the results presented in this paper are not publicly available at this time but may be obtained from the authors upon reasonable request.

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
