# Peer review of "A 1-μm-Band Injection-Locked Semiconductor Laser with a High Side-Mode Suppression Ratio and Narrow Linewidth"

_sensors, 2022, doi:10.3390/s22239239_

Round 1
Reviewer 1 Report
The construction of the device and the results are interesting. As authors said in conclusions, such device could find applications in sensors etc.. In fact the device is not a sensor - I'm not sure if the "sensors" journal is the best journal where such results should be published.
Despite this the paper can be published after minor corrections:
1. The using of fiber gratting to force single-frequency operation of semiconductor laser/diode is not new (e.g. https://doi.org/10.1364/OE.24.008391 ) , but there is no reference for any paper which describe such construction. In my opinion this should be improved.
2. The difference between picture Fig.1b and Fig.1c is clearly, but the origin of there pictures are marked not very well. Why these pictures are different if they shows the same part of fiber? It is not clear.
3. In my opinion, In Fig.2a the peaks should be marked by central wavelength for clarity.
4. There are also small mistakes in the text:
Line 94: there is "Fabry-Pero" - should be "Fabry-Péro"
Line 96: "The light entering..., making the light of this mode more competetive" - in my opinion in this sentence there is missing information that the FBG has narrow reflection characteristic and that is why the light of this mode is more competetive.
Line 98: "The carrier in the cavity changes..." is it correct that there is only carrier? not carrier density or distribution?
Line 157: "...butterfly shape..." I suggest change to "...butterfly housing..."
Line 161, 162, and 164: there should be "ILX Lightwave" insted of "ILX light wave"
Line 163: in case of teperature units there should be no space between number and unit, so no "20 oC", but "20oC"
Line 199: the title of point "3.3" should be started by capital letter
Line 211: "...so the power level is close." In my opinion this should be expleined
Line 220: to many spaces before "characteristics"
Line 222: "...of linewidth under complete Fourier..." it is not clearly for me, in my opinion also this should be explained
Reviewer 2 Report
Based on the external optical feedback injection locking technology with a quantum dot gain chip and a Fs-apodized FBG, a single frequency output is achieved in a 1-μm band. The maximum SMSR is 66.3 dB, and the maximum output power is 134.6 mW. The measured linewidth is 260.5 kHz. Meanwhile, a minimum integrated linewidth of less than 180.4 kHz was observed. It will contribute to the high-power fiber laser seed source for the spectral beam combination and multi-stage energy amplification. However, there are some problems as follows:
1. As the front surface anti-reflectivity of the gain chip has a direct impact on the characteristics of the external cavity laser lasers. In line 141, the authors describe the value of the front surface anti-reflectivity (AR) coating of the gain chip is less than 0.01%. Then, is this value of the front surface reflectivity come from the actual measurement, and how to measure this reflectivity with the value of 0.01%? How many layers in the anti-reflectivity film, what are the materials, and what about the tolerances. The authors are recommended to add this information further.
2. Figure 3 gives the excitation spectrum. But the SMSR values described in the text of line 170 cannot be observed in the visual reflection in figure 3, the authors are recommended to complete figures and make those values correspond to the text.
3. In line 172, “the spectrum at 50 mA shows the peak transmission of FBG, which is basically consistent with Fig. 2c”. How to understand this sentence? Is it a pen error by the authors, Figure 2c should be changed to Figure 2b?
4. The English expression is not authentic enough. For example, in line 36, “…achieve the highest possible side-mode suppression ratio (SMSR)…”is suggested to be changed into “…achieve the side-mode suppression ratio (SMSR) as high as possible…” For another example, in line 42, “…commonly used schemes to achieve 1-μm band lasers…”is suggested to be changed into "…the schemes commonly used to achieve 1-μm band lasers… " etc.
